# ADVANCING FORMAL MATHEMATICAL REASONING WITH EXPLORATIVE REINFORCEMENT LEARNING

## ABSTRACT

Reinforcement learning with verifiable rewards is a promising direction for training large language models (LLMs) in formal reasoning. However, current approaches such as GRPO and expert iteration, which generate multiple solution candidates per problem and assign Pass@1 rewards to each candidate independently, struggle to balance exploration and exploitation, and thus fail to acquire new proving patterns (*e.g.*, proof by contradiction, case analysis, mathematical induction). The resulting Pass@1 RL learned policies tend to over-rely on conservative actions (*e.g.*, workaround Lean 4 proof completion by `sorry`) inherited from pretraining and supervised fine-tuning (SFT), thereby reinforcing misplaced confidence in these shortcuts during inference-time scaling. To address this limitation, we introduce T-RL, the first exploration-aware RL method in formal reasoning that leverages compiler rewards aligned with the Pass@K effect to enhance grouped Lean4 proof completion and self-improvement directly. Empirically, T-RL improves exploration by increasing the average number of tactics per proof and by encouraging the use of more diverse mathematical techniques. Our T-RL–trained prover, Qwen2.5-1.5B, outperforms DeepSeek-Prover-V1.5-7B on both MiniF2F and FormalMATH-Lite. Specifically, it achieves $70.1\%$ on MiniF2F with only $1 \times 32 \times 4$ sampling budgets higher than Deepseek-V1.5-RL with Pass@$16 \times 6400$ by MCTS. T-RL is a primary reinforcement learning algorithm with explicit exploration-based learning objectives, demonstrating promising preliminary results and highlighting a potential direction for future research in formal reasoning.

## 1 INTRODUCTION

Recent advancements in formal mathematical reasoning within large language models (Wang et al., 2025) demonstrate that performance can be enhanced through training on extensive reasoning traces (Ren et al., 2025) that were distilled from stronger models with complex chain-of-thought reasoning patterns (Wei et al., 2022) These methods include online reinforcement learning (RL) (Ren et al., 2025) or offline expert iteration (Lin et al., 2025). However, prior studies, such as Wu et al. (2024), indicate that both RL and expert iteration approaches tend to saturate at low pass rates at early training stages. This saturation occurs because the probability of generating a new correct proof diminishes exponentially as the complexity of the theorem increases. Furthermore, recent research (Yu et al., 2025b) highlights that test-time scaling strategies, including Best-of-$N$ sampling and best-first search (Xin et al., 2025), yield only marginal improvements. These methods often expend significant computational resources on generating incorrect proofs, which fail to provide meaningful training signals.

A critical bottleneck in these approaches lies in the learning objective. Standard RL methods, such as GRPO (Guo et al., 2025), primarily optimize Pass@1, leading to a rapid collapse of the policy's probability mass onto a narrow set of high-likelihood actions. This phenomenon frequently inherits conservative shortcuts from supervised fine-tuning (SFT) from the distilled dataset from stronger models (Yu et al., 2024), thereby reducing generation diversity.

We argue that RL algorithms for formal mathematical reasoning should prioritize exploration as a central component for overcoming current limitations. In this domain, exploration is particularly critical yet inherently difficult, since the diversity of valid Lean4 programs is restricted and

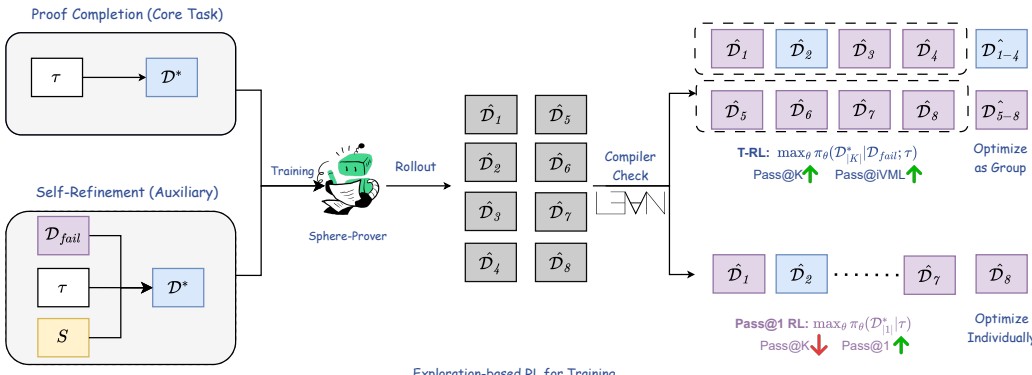

Figure 1: An overview of the proposed explorative RL algorithm (T-RL) for formal reasoning. T-RL directly optimizes both Pass@$K$ and tree search, while accounting for the group effects of proof completion (core task) and self-refinement (an auxiliary task). In contrast, Pass@1 rewards each proof in isolation and therefore lacks the ability to balance exploration and exploitation.

the amount of available data remains limited. Therefore, how to encourage the exploration in formal mathematical reasoning during the RL training becomes quite important. The Pass@$K$ objective (Walder & Karkhanis, 2025; Tang et al., 2025) provides a suitable measure for this goal, which emphasizes the success with a group of responses, and would reward positively to the failed proofs with high-quality intermediate proving procedures and excellent usage of tactics or mathematical skills (*e.g.*, proof by contradiction). However, the application of Pass@$K$ RL methods, predominantly developed for natural-language reasoning, remains underexplored in the context of formal reasoning, where the long-context window and symbolic feedback impose unique challenges.

To bridge these gaps, we propose T-RL, to our knowledge the first exploration-driven RL framework for formal reasoning, which integrates GRPO (Shao et al., 2024) with the Pass@$K$ inference strategy for Lean4 proof generation. We also incorporate auxiliary task training with self-refinement objectives, enabling provers to better interpret mixed-symbolic formal contexts semantically and refine flawed proofs using compiler feedback, supported by a group-based optimization strategy for parallel best-of-refinement. As shown in Section 3, T-RL-trained provers outperform Pass@1 RL counterparts on all challenging Lean4 benchmarks. For example, Qwen2.5-1.5B (T-RL) achieves 56.6% with Pass@32, exceeding 53.6% for Qwen2.5-1.5B (GRPO) on MiniF2F, while sustaining higher token-level entropy, indicating enhanced exploration. Additionally, T-RL fosters diverse proof strategies; the T-RL-trained Goedel-Prover (Lin et al., 2025) surpasses baselines and GRPO under identical sampling budgets across all benchmarks, and mastering advanced skills like proof by contradiction and problem decomposition while minimizing reliance on `sorry` workarounds. In summary, the key contributions of this work are as follows:

- **Explorative Reinforcement Learning for Lean4**: We present T-RL, the first explorative RL algorithm for formal reasoning, enhancing the Pass@$K$ proof completion and self-refinement capabilities of LLM-based provers.

- **T-RL Outperforms GRPO in Exploration**: T-RL surpasses GRPO, the predominant RL algorithm for provers, across multiple exploration aspects. It maintains consistently higher token-level entropy, fosters diverse proof strategies, and improves overall reasoning ability (ARA) and reliability (RRA), opening a promising new direction for RL in formal reasoning.

- **Provers with Competitive Performance**: Our T-RL-trained provers, *e.g.*, Qwen2.5-1.5B, outperform larger models, *e.g.*, DeepSeek-Prover-V1.5, achieving 70.1% with Pass@$1 \times 4 \times 32$ on MiniF2F compared to 62.7% with Pass@$16 \times 6400$ sampling budgets.

## 2 EXPLORATIVE REINFORCEMENT LEARNING FOR FORMAL REASONING

Figure 1 illustrates the overall pipeline of our proposed reinforcement learning approaches for formal reasoning. We describe the details of the proposed explorative RL (T-RL) in Section 2.1.

## 2.1 POLICY TRAINING WITH EXPLORATION

**Reward Modeling.** We design a rule-based reward system composed of two components. 1. Accuracy reward $\mathcal{F}_i \in \{0, 1\}$ given by Lean4 compiler: a binary signal indicating the correctness of the whole final proof, weighted by $\alpha = 0.9$. 2. Template reward $t_i \in \{0, 1\}$: a structural constraint requiring the model to output both natural-language reasoning traces (thinking steps) and the final Lean4 proof, weighted by $\beta = 0.1$. The overall reward for a proof trajectory is thus defined as:

$$g_i(d; \tau) = \alpha \mathcal{F}_i(d; \tau) + \beta t_i(d; \tau). \tag{1}$$

**RL with Auxiliary Task Training.** 1. Core task (proof completion): given a theorem statement $\tau$, the model generates a complete formal proof $\mathcal{D}$. 2. Auxiliary task (self-refinement): given a flawed proof $\mathcal{D}_i$, the statement $\tau$, and its corresponding symbolic proof state $S$ provided by the Lean4 compiler, which captures the error feedback of the flawed tactics as well as the corresponding evolution of proof goal states, the model is tasked with revising $\mathcal{D}_i$ into a correct proof $\mathcal{D}_{i+1}^*$. The overall optimiazation objective for multitask RL is to strengthen the core task while injecting the capability to perform the auxiliary task:

| Method | MiniF2F | FormalMATH-Lite | Proverbench |
|---|---|---|---|
| Vanilla | 45.9 | 33.6 | 3.6 |
| SFT | 52.0 | 40.9 | 5.0 |
| Multi-task SFT | 48.8 | 36.9 | **7.0** |
| GRPO (Zero) | 46.7 | 36.0 | 3.3 |
| GRPO (Initialized) | 53.6 | 41.6 | 3.6 |
| T-RL (Zero) | 47.1 | 36.0 | 3.6 |
| T-RL (Initialized) | **56.6** | **41.6** | 5.0 |

Table 1: Performance comparison of Qwen2.5-1.5B-10K across different learning methods, measured by Pass@32 on Lean4 benchmarks.

$$\arg \max_\theta \ J_\pi \left[ g(d; \tau) \right] + \lambda_r J_{\pi_r} \left[ g(d; s; d_{fail}; \tau) \right]. \tag{2}$$

$g(d; s; d_{\text{fail}}; \tau)$ in Equation (2) represents a special case of Equation (1), sharing the same reward mechanism for evaluating Pass@1 self-refinement correctness.

**Exploration-based RL.** Our goal is to maximize a continuous relaxation of the Pass@$K$ metric for formal reasoning with the training objective in activate iVML. Formally, we rewrite Equation (2) with awareness of test-time scaling:

$$J(\theta) = \text{Pass@K}(\theta) = \mathbb{E}_{\tau \sim Ds, \{\hat{d}_i\}_{i=1}^k \sim \pi_\theta(\cdot|\tau)} \left[ \max \left( g_1(\hat{d}_1; s_1; d_{fail_1}; \tau), \ldots, g(\hat{d}_k; s; d_{fail_k}; \tau) \right) \right] \tag{3}$$

At each training step, the policy $\pi_\theta$ draws $n \geq K$ i.i.d. rollouts $\hat{d}_i \sim \pi_\theta(\cdot|\tau)$. Rewards are computed using Equation (1) and then sorted. An unbiased policy-gradient estimator for Equation (3) is:

$$\hat{\nabla} J(\theta) = \sum_{i=1}^n s_i \nabla_\theta \log \pi_\theta(\hat{d}_i|\tau) \tag{4}$$

$$s_i = S(i, K, [n]) \equiv \frac{1}{\binom{n}{K}} \sum_{\substack{I \subseteq [n], |I| = K, \\ i \in I}} \max_{j \in I} g_{(j)}, \qquad [n] \equiv \{1, \ldots, n\}. \tag{5}$$

However, directly using Equation (4) often leads to high variance, we thus subtract the transformed reward by a leave-one-out (LOO) baseline (Walder & Karkhanis, 2025):

$$s_{(i)}^{\text{loo}} = S(i, K, [n]) - \frac{1}{n-1} \sum_{j \neq i}^n S(j, K, [n] \setminus \{i\}). \tag{6}$$

We adapt Equation (6) with GRPO to enable tree search awareness (*i.e.*, T-RL):

$$\mathcal{L}_{TRL}(\theta) = -\frac{1}{N} \sum_{i=1}^n \left( \min \left( \frac{\pi_\theta(\hat{d}_i \mid \tau)}{\pi_{\text{old}}(\hat{d}_i \mid \tau)} s_i^{\text{loo}}, clip \left( \frac{\pi_\theta(\hat{d}_i \mid \tau)}{\pi_{\text{old}}(\hat{d}_i \mid \tau)}, 1 - \epsilon, 1 + \epsilon \right) s_i^{\text{loo}} \right) \right) - \beta \text{KL} \left( \pi_\theta \| \pi_{\text{ref}} \right) \tag{7}$$

In Section 3, we find that T-RL not just enhances the overall performance of provers on existing Lean4 benchmarks, but also consistently maintains higher entropy than GRPO, indicating its superior exploration capabilities in formal reasoning.

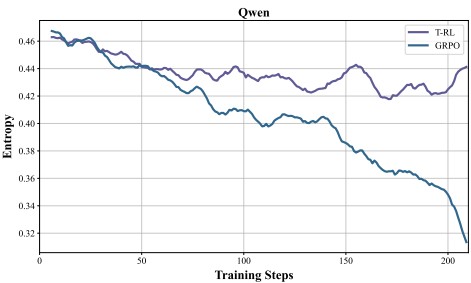 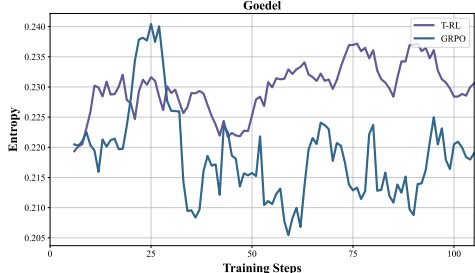

Figure 2: Training Entropy: T-RL vs. GRPO on Qwen2.5-1.5B and Goedel provers.

## 3 DISCUSSION AND INTRIGUING INSIGHTS IN THE PERSPECTIVE OF EXPLORATION

**T-RL outperforms GRPO in Token-level Exploration.** The Pass@32 results of T-RL–trained provers (Goedel-24K in Table 2 and Qwen2.5-1.5B-10K in Table 1) demonstrate superior performance compared to GRPO on challenging Lean4 benchmarks (MiniF2F, FormalMATH-Lite, and Proverbench). Under the Goedel-24K training setup, T-RL achieves 80.7 on MiniF2F, 49.2 on FormalMATH-Lite, and 34.2 on Proverbench (nearly three times its

| Method | MiniF2F | FormalMATH-Lite | Proverbench |
|--------|---------|-----------------|-------------|
| Vanilla | 78.3 | 48.0 | 13.2 |
| GRPO | 79.0 | 48.0 | 12.6 |
| T-RL | **80.7** | **49.2** | **34.2** |

Table 2: Performance comparison of Goedel-prover across different training methods, measured by Pass@32.

GRPO counterpart), whereas GRPO attains 79.0, 48.0, and 12.6, respectively. For Qwen2.5-1.5B, T-RL consistently outperforms GRPO across all benchmarks under both the zero-shot setting (without distillation SFT) and the initialized setting (with distillation SFT). For instance, on MiniF2F, T-RL (Zero) reaches 47.1 compared to 46.7 for GRPO (Zero), and T-RL (Initialized) achieves 56.6 compared to 53.5 for GRPO (Initialized). Except for the evaluation performance, we notice that T-RL consistently maintains higher token-level entropy than GRPO (See Figure 2). For example, Qwen2.5-1.5B with GRPO exhibits a consistently decreasing entropy throughout training, whereas T-RL maintains entropy above 0.42. According to (Cui et al., 2025), LLMs often trade entropy for higher test performance, following:

$$R = -\alpha \exp(\mathcal{H}) + b \tag{8}$$

where $\mathcal{H}$ denotes the token-level entropy. These results indicate that T-RL can balance evaluation performance and entropy more effectively, outperforming GRPO in maintaining a better trade-off between exploitation and exploration during training.

**T-RL as a Stronger Multi-Task Learner.** Table 1 shows that multi-task SFT provides only limited gains: while it improves Proverbench (7.0 vs. 5.0), it hurts MiniF2F and FormalMATH-Lite, indicating overfitting to the dominant dataset. In contrast, T-RL consistently improves performance. With initialization, it reaches 56.6 on MiniF2F (+3.0 over GRPO, +4.6 over SFT) and matches the best supervised baseline on FormalMATH-Lite (41.6).

| Tactics | Goedel | Goedel (GRPO) | Goedel (T-RL) |
|---------|--------|---------------|---------------|
| **by_contra** (↑) | 3538 | 3722 | **4049** |
| **contradiction** (↑) | 4876 | 5372 | **6744** |
| **induction** (↑) | 546 | 508 | **592** |
| **rcases** (↑) | 2593 | 2756 | **2936** |
| **sorry** (↓) | 4302 | 4052 | **3536** |

Table 3: Comparison of the tactics used by Goedel Prover on FormalMATH-Lite Benchmark via Pass@32 (425 Problems).

**T-RL Outperforms GRPO in Exploring Diverse Proof Strategies.** In Lean4, the choice of tactics reflects different proof strategies. To examine this, we compare the average number of tactics used per problem (Table 5) and the distribution of tactic preferences (Table 3). For instance, employing `by_contra` or `contradiction` indicates proof by contradiction, `induction` corresponds to induction-based reasoning, and `rcases` reflects problem decomposition. By contrast, `sorry` typically signals failure to complete a proof. From Table 3, we observe that T-RL not only employs

more diverse tactics on average but also shows clear shifts in tactic selection, demonstrating greater strategic diversity. On FormalMATH, T-RL uses proof by contradiction more often (4049 vs. 3538), applies the `contradiction` tactic more frequently (6744 vs. 4876), and employs induction more often (592 vs. 546) than the baseline Goedel-Prover. At the same time, reliance on `sorry` (which is a workaround of proof completion) is substantially reduced (3536 vs. 4302), indicating that T-RL encourages the prover to pursue complete proofs rather than abandon answering. The baseline tends to solve the problem in a calculation form using the known lemma $\sum_{k=0}^{n-1} 2^k = 2^n - 1$, applied via `rw [Nat.sub_eq_of_eq_add] <;> rfl`. The theorem $\sum_{k \in \text{Finset.range } 101} 2^k \mod 7 = 3$ is proven using distinct strategies by Goedel-Prover and T-RL trained Goedel-Prover. Our model Proves a generalized form $\forall n : \mathbb{N}, \sum_{k \in \text{Finset.range } n} 2^k = 2^n - 1$ using structural induction. Employs tactics like `simp`, `rw [Finset.sum_range_succ, pow_succ]`, `ring_nf`, and `omega` to handle base and inductive cases, resembling step-by-step derivation, showing an explored strategy to takle more challenging problem.

---

**Example: Comparing Proof Strategies of T-RL trained Goedel-Prover with baseline.**

```
-- Proof generated by Goedel−Prover, relying solely on known theorems.
theorem mathd_numbertheory_127_Goedel : (Σ k in Finset.range 101, 2 ^ k) % 7 = 3 := by
 have h_sum : (Σ k in Finset.range 101, 2 ^ k) = 2 ^ 101 − 1 := by
  have h₁ : (Σ k in Finset.range 101, 2 ^ k) = 2 ^ 101 − 1 := by
   rw [Nat.sub_eq_of_eq_add]
   <;> rfl

-- Proof generated by T−RL trained goedel−prover, employing an induction strategy to tackle a more challenging proposition.
theorem mathd_numbertheory_127_trl : (Σ k in Finset.range 101, 2 ^ k) % 7 = 3 := by
 have h_sum : (Σ k in Finset.range 101, 2 ^ k) = 2 ^ 101 − 1 := by
  have h : ∀ n : ℕ, (Σ k in Finset.range n, 2 ^ k) = 2 ^ n − 1 := by
   intro n
   rw [Nat.sub_eq_of_eq_add]
   induction n with
   | zero => simp
   | succ n ih =>
    cases n with
    | zero => simp [Finset.sum_range_succ]
    | succ n =>
     simp_all [Finset.sum_range_succ, pow_succ, mul_add, mul_one, Nat.mul_sub_left_distrib]
    <;> ring_nf at *
    <;> omega
```

## 4 EXPERIMENTS AND RESULTS

This section presents our experiment results on T-RL trained provers series on Qwen2.5-1.5B and Goedel-Prover-V2. We first introduce our evaluation settings in Section 4.1, then discuss the main evaluation results and qualitative analysis on the benchmarks in Section 4.2.

### 4.1 IMPLEMENTATION DETAILS

**Training Strategies and Datasets.** We train Qwen2.5-1.5B in two stages: SFT followed by T-RL. We curate 30K statements from Numina-Lean Wang et al. (2025) and augment with CoT distilled via expert iteration from DeepSeek-Prover-V2. For T-RL, we use 4K statements for proof generation and 1K flawed proofs for the proof refinement task. For Goedel-Prover-V2, we adopt continual T-RL without SFT, build a 5K multitask RL dataset.

**Hyperparameters.** For T-RL training, we use a temperature of 1.0 for all models. For Goedel-Prover, we set the maximum context length to 24,000 tokens, allocating 4,000 tokens for the prompt and 20,000 for the response. For Qwen2.5-1.5B, we use a maximum context length of 10,096 tokens, with 2,000 tokens for the prompt and 8,096 for the response. We employ a mini-batch size of $n = 32$ and optimize for Pass@$K$ with $K = 4$, chosen as a balanced value under our computational constraints. For evaluation, we set the temperature to 0.7 for all models. For Qwen2.5-1.5B, we use a maximum context length of 10,096 tokens. Although Goedel-Prover originally supports a 32k context window, we uniformly adopt a maximum of 24k tokens for both training and evaluation to ensure a fair comparison. All experiments are conducted on a cluster of 8 H100 GPUs. We use the 4.15.0 version with a timeout of 100 seconds for the Lean4 configuration.

| Method | Model size | Sample budget | FormalMATH-Lite | MiniF2F |
|---|---|---|---|---|
| *Tree Search Methods* | | | | |
| BFS-Prover | 7B | $32 \times 32 \times 100$ | 45.9% | – |
| | | $2048 \times 2 \times 600$ | – | 70.8% |
| InternLM2.5-StepProver + BFS | 7B | $32 \times 32 \times 100$ | 25.7% | – |
| | | $256 \times 32 \times 600$ | – | 65.9% |
| *Whole-proof Generation Methods* | | | | |
| Goedel-Prover-V1 | 7B | 32 | 46.7% | 57.6% |
| STP | 7B | 32 | 48.6% | – |
| | | 128 | 50.4% | 57.2% |
| DeepSeek-Prover-V1.5-SFT | 7B | 32 | 40.4% | 48.2% |
| | | 128 | 42.1% | 51.6% |
| | | 1024 | 45.1% | – |
| DeepSeek-Prover-V1.5-RL | 7B | 32 | 48.0% | 50.0% |
| | | 128 | 48.8% | – |
| | | 1024 | 49.7% | – |
| | | 3200 | – | 54.9% |
| | | $4 \times 6400$ | – | 58.4% |
| DeepSeek-Prover-V2 | 7B | 1 | 51.7% | 58.6% |
| | | 32 | 53.4% | 72.2% |
| | | 1024 | **54.1%** | – |
| Goedel-Prover-V2-24K | 8B | 32 | 48.0% | 78.3% |
| Qwen2.5-1.5B-10K (T-RL) | 1.5B | 32 | 41.6% | 56.6% |
| | | 128 | 43.5% | 59.0% |
| | | $1 \times 32 \times 4$ | 45.4% | 70.1% |
| Goedel-Prover-V2-24K (T-RL) | 8B | 32 | 49.2% | 80.7% |
| | | 128 | 52.0% | 82.4% |
| | | $1 \times 32 \times 4$ | 53.4% | **83.6%** |

Table 4: Comparison with state-of-the-art models on the FormalMATH-Lite and MiniF2F dataset.

## 4.2 MAIN RESULTS

**Competitive Performance of T-RL Trained Provers on MiniF2F.** Table 4 presents a comparative analysis of several theorem-proving methods on the miniF2F-test dataset. Qwen2.5-1.5B (T-RL) achieves a pass rate of 56.6% with Pass@32, outperforming DeepSeek-Prover-V1.5-RL, which reaches 54.9% with Pass@3200, and only slightly below its performance of 58.4% with Pass@4 × 6400 using MCTS. Leveraging iVML Yu et al. (2025c) with far lower computational cost than the MCTS employed by DeepSeek-Prover-V1.5, Qwen2.5-1.5B (T-RL) achieves 70.1%, surpassing InternLM2.5 with BFS (65.9% with Pass@256 × 32 × 600), and is close to Deepseek-Prover-V2 with Pass@32 (72.2%). It is worth mentioning that DeepSeek-Prover-V2 is a 7B model trained with heavy expert iteration on synthetic data, while Qwen2.5-1.5B (T-RL) only leverages 30K distilled CoT for initialization, the closed results showing the effectiveness of the T-RL and iVML methods. Goedel-Prover-V2 (T-RL) attains a pass rate of 80.7% with Pass@32 under a 24K context window, exceeding the 78.3% Pass@32 performance of its base model trained with expert iteration, DAPO, and model merging. With iVML at Pass@1 × 32 × 4, Goedel-Prover-V2 (T-RL) further achieves 83.2% on miniF2F, demonstrating that iVML effectively enables exploration for solving more challenging problems at test time.

**T-RL Improves the Provers on Harder Benchmarks.** Table 4 compares T-RL methods with existing theorem provers on the FormalMATH-Lite benchmark. FormalMATH-Lite is a particularly challenging Lean4 dataset, spanning problems from high-school Olympiad competitions to undergraduate-level theorems across diverse domains such as algebra, calculus, number theory, and discrete mathematics. Qwen2.5-1.5B (T-RL) achieves a pass rate of 41.6% with Pass@32, already surpassing DeepSeek-Prover-V1.5-SFT (40.4%) at the same budget. With iVML search ($1 \times 32 \times 4$),

| Models | Goedel | Goedel(GRPO) | Goedel(T-RL) | Qwen | Qwen(GRPO) | Qwen(T-RL) |
|---|---|---|---|---|---|---|
| *FormalMATH-Lite* | | | | | | |
| Average Tactic Count | 4.9 | 4.8 | **5.5** | 4.1 | 4.6 | **4.8** |
| ARA (%) | 36.8 | 37.3 | **38.1** | 23.4 | 26.8 | **28.7** |
| RRA (%) | 30.1 | 31.0 | **31.2** | 14.5 | 18.5 | **20.7** |
| *MiniF2F* | | | | | | |
| Average Tactic Count | 6.0 | **6.99** | 6.4 | 4.1 | 4.3 | **5.0** |
| ARA (%) | 57.4 | 57.3 | **59.1** | 25.2 | 27.2 | **36.3** |
| RRA (%) | 45.9 | 49.5 | **50.0** | 15.1 | 15.1 | **27.8** |
| *ProverBench* | | | | | | |
| Average Tactic Count | 3.8 | 3.9 | **4.7** | 1.5 | 1.9 | **2.4** |
| ARA (%) | 6.8 | 7.0 | **10.0** | 1.5 | 1.9 | **2.5** |
| RRA (%) | 3.3 | 4.0 | **6.1** | 0.3 | 0.9 | **1.5** |

Table 5: Overall Performance comparison accross different learning methods.

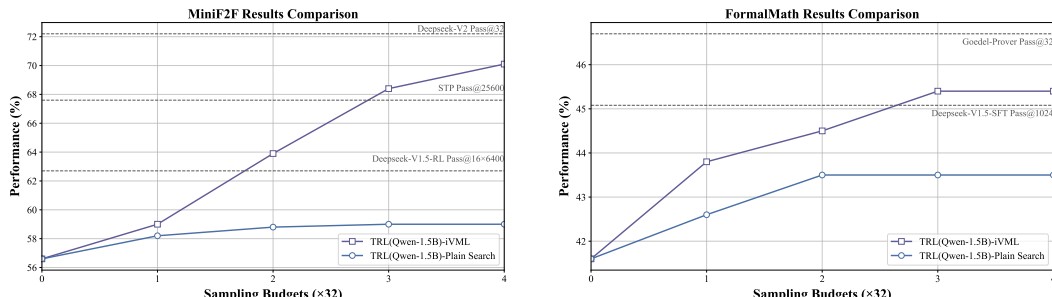

Figure 3: Comparison of Qwen2.5-1.5B with different test-time scaling approach on Lean4 Benchmarks.

it further improves to $45.4\%$, approaching BFS-Prover ($45.9\%$) with $32 \times 32 \times 100$ sampling budgets. This demonstrates that T-RL combined with lightweight iVML enables competitive results even with much smaller model sizes and significantly lower sample budgets than BFS approaches.

**T-RL Improves the Overall Performance of Theorem Proving.** While raw pass rates (*e.g.*, Pass@$K$) are commonly used to evaluate provers, they provide only a limited view of model capability. A high overall pass rate may hide important differences in problem-level behavior: a prover might solve many easy problems but fail consistently on harder ones, or succeed on some tasks but remain unreliable across trials. To capture these finer aspects of reasoning performance, we introduce two complementary metrics. *1. Average Reasoning Ability (ARA)* measures a prover's overall success rate across all problems under Pass@32:

$$\text{ARA} = \frac{1}{N} \sum_{i=1}^{N} \text{PassRate}_i, \tag{9}$$

where $N$ is the number of problems and $\text{PassRate}_i$ is the pass rate for the $i$-th problem. This metric reflects the prover's average problem-solving ability. *2. Reliable Reasoning Ability (RRA)* complements ARA by quantifying consistency: it measures the proportion of problems for which the prover achieves a pass rate above $80\%$ under Pass@32:

$$\text{RRA} = \frac{1}{N} \sum_{i=1}^{N} \mathbf{1}(\text{PassRate}_i > 0.8) \tag{10}$$

As shown in Table 5, T-RL consistently improves over both supervised training and GRPO. On FormalMATH-Lite, Goedel(T-RL) slightly outperforms Goedel(GRPO) in both ARA (38.1% vs. 37.3%) and RRA (31.2% vs. 31.0%), while Qwen2.5-1.5B (T-RL) shows larger margins (ARA 28.7% vs. 26.8%, RRA 20.7% vs. 18.5%). On MiniF2F, T-RL again outperforms GRPO, with Qwen2.5-1.5B (T-RL) achieving particularly strong gains (ARA 36.3% vs. 27.2%, RRA 27.8% vs.

15.1%). Even on the low-success regime of ProverBench, T-RL surpasses GRPO, *e.g.*, Goedel(T-RL) improves RRA from 4.0% to 6.1%. These results confirm that T-RL offers more robust and reliable improvements than GRPO, especially for Qwen2.5-1.5B.

---

**Example: Comparing proof generated by plain sampling with proof generated by self-refinement.**

Incorrect proof generated by Best-of-N search:

```
theorem algebra_bleqa_apbon2msqrtableqambsqon8b (a b : ℝ) (h₀ : 0 < a ∧ 0 < b) (h₁ : b ≤ a) :
  (a + b) / 2 − Real.sqrt (a * b) ≤ (a − b)^2 / (8 * b) := by
 have h_main : (a + b) / 2 − Real.sqrt (a * b) ≤ (a − b)^2 / (8 * b) := by
  have h₂ : 0 < b := by linarith
  have h₃ : 0 < a := by linarith
  have h₄ : 0 < a * b := by positivity
  have h₅ : 0 ≤ Real.sqrt (a * b) := Real.sqrt_nonneg (a * b)
  have h₆ : 0 < 8 * b := by positivity
  have h₇ : 0 ≤ (a − b)^2 := by positivity
  have h₈ : 0 ≤ (a − b)^2 / (8 * b) := by positivity
  have h₉ : 0 ≤ Real.sqrt (a * b) := by positivity
  nlinarith [sq_nonneg (Real.sqrt (a * b) − (a + b) / 2), sq_nonneg (Real.sqrt (a * b) − (a − b) / 2),
   Real.sq_sqrt (show 0 ≤ a * b by positivity),
   sq_nonneg (a − b), sq_nonneg (Real.sqrt (a * b) − (a + b) / 2),
   sq_nonneg (Real.sqrt (a * b) − (a − b) / 2), sq_nonneg (a − b),
   sq_nonneg (Real.sqrt (a * b) − 1), sq_nonneg (Real.sqrt (a * b) − 2)]
 exact h_main
```

Error Information of the flawed proof:

```
a b : ℝ
h₀ : 0 < a ∧ 0 < b
h₁ : b ≤ a
h₂ : 0 < b
...
h₉ : 0 ≤ \sqrt{(a * b)}
a+ : (a + b) / 2 − √(a * b) > (a − b)^2 / (8 * b)
⊢ False
linarith failed to find a contradiction
```

Correct proof generated by self-refinement of T-RL trained prover with error feedback.

```
theorem algebra_bleqa_apbon2msqrtableqambsqon8b (a b : ℝ) (h₀ : 0 < a ∧ 0 < b) (h₁ : b ≤ a) :
  (a + b) / 2 − Real.sqrt (a * b) ≤ (a − b)^2 / (8 * b) := by
 have h_main : (a + b) / 2 − Real.sqrt (a * b) ≤ (a − b)^2 / (8 * b) := by
  have h₂ : 0 < a * b := by nlinarith
  have h₃ : 0 < Real.sqrt (a * b) := Real.sqrt_pos.mpr h₂
  have h₄ : 0 < 8 * b := by nlinarith
  have h₅ : 0 < Real.sqrt (a * b) * (8 * b) := by positivity
  field_simp [h₄.ne']
  rw [div_le_div_iff (by positivity) (by positivity)]
  nlinarith [sq_nonneg (a − b), Real.sq_sqrt (show 0 ≤ a * b by nlinarith),
   sq_nonneg (a + b − 2 * Real.sqrt (a * b)),
   sq_nonneg (a − b + 2 * Real.sqrt (a * b)),
   sq_nonneg (a + b + 2 * Real.sqrt (a * b)),
   sq_nonneg (a − b − 2 * Real.sqrt (a * b)),
   sq_nonneg (a + b − Real.sqrt (a * b)),
   sq_nonneg (a − b + Real.sqrt (a * b)),
   sq_nonneg (a + b + Real.sqrt (a * b)),
   sq_nonneg (a − b − Real.sqrt (a * b))]
 exact h_main
```

---

**T-RL Improves Familiarity of Using Tactics.** Table 5 shows that T-RL not only raises ARA and RRA, but also increases the average tactic count across benchmarks, indicating that models learn to deploy these tactics more effectively. For example, Goedel(T-RL) generates 5.5 tactics on FormalMATH-Lite compared to 4.8 with GRPO, while Qwen(T-RL) expands from 4.6 to 4.8. On MiniF2F, Qwen(T-RL) shows the largest jump, from 4.3 to 5.0. These longer and more structured tactic sequences suggest that T-RL improves the model's familiarity and flexibility in combining tactics, leading to more reliable reasoning strategies. The following list summarizes a set of commonly used Lean tactics that frequently appear in proof generation and are what we observed in this experiment. These tactics cover a variety of proof styles, including algebraic simplification (`ring_nf`, `norm_num`, `field_simp`), case analysis (`cases`, `induction`, `by_cases`), rewriting (`rw`, `simp`, `unfold`), and structural reasoning (`have`, `apply`, `constructor`, `by_contra`).

## 4.3 TEST-TIME SCALING OF T-RL

**Evaluation Settings.** In this section, we evaluate the T-RL–trained Qwen2.5-1.5B using different test-time scaling approaches to observe their performance with larger sampling budgets: 1. Best-of-N plain search: BoN, which treats each proof candidate as independently and identically distributed (i.i.d.) and evaluates performance using the Pass@$K$ success rate. 2. iVML: Self-refinement with instance-verbalized machine learning (iVML) Yu et al. (2025c), which differs from Goedel Lin et al. (2025) in that we apply refinement under a best-of-N setting. Specifically, instead of refining individual proofs, we iteratively refine groups of proofs based on error feedback from the previous iteration, and then evaluate their Pass@$K$ performance in the same manner as Best-of-N sampling.

**Results.** Figure 3 presents the comparison between plain Best-of-N search and iVML across MiniF2F and FormalMath benchmarks. On MiniF2F, iVML demonstrates a clear scaling advantage: as the sampling budget increases, its performance improves steadily from 56.5% to 70.1%, while plain search saturates around 59.0%. Remarkably, iVML at a budget of $4 \times 32$ achieves performance better than much larger models with substantially higher sampling costs, such as DeepSeek-V1.5-RL and STP. On FormalMath, iVML also consistently outperforms plain search, though the gains are less pronounced than on MiniF2F. Performance increases from 41.5% to 45.5% with iVML, surpassing DeepSeek-V1.5-SFT but remaining slightly below the Goedel-Prover. These results highlight that iVML not only improves sample efficiency by leveraging error-driven refinement but also scales more effectively under larger sampling budgets. Through self-refinement during exploration-based RL, iVML activates the capabilities of T-RL beyond the limits of plain search, enabling higher performance.

**Qualitative Analysis.** In the following example, the proof generated by plain search fails due to redundant or misaligned auxiliary lemmas, and `linarith` is unable to derive a contradiction. In contrast, iVML leverages error feedback to iteratively refine groups of proofs, allowing the prover to correct earlier mistakes. By explicitly recognizing where the initial proof attempt fails (e.g., the inequality direction error and the failed `linarith` call), the refinement step generates a corrected proof strategy that introduces more targeted auxiliary lemmas and rewrites. For instance, in the refined proof, error feedback leads to the use of `div_le_div_iff` and carefully structured `nlinarith` calls, which resolve the inequality reasoning properly.

## 5 RELATED WORK AND CONCLUDING REMARKS

**Formal Mathematical Reasoning.** Recent LLM-based provers primarily rely on extensive test-time scaling strategies (*e.g.*, expert iteration (Polu & Sutskever, 2020; Anthony et al., 2017; Lin et al., 2025)), followed by Pass@1 reinforcement learning methods (*e.g.*, GRPO (Ren et al., 2025; Xin et al., 2024; Wang et al., 2025)) to further enhance formal reasoning capabilities (Yu et al., 2025b; Yang, 2023). Recent approaches (Xin et al., 2024; Yu et al., 2025c; Lin et al., 2025) have introduced self-improvement (an exploitation-oriented method) (Yao et al., 2023; Yu et al., 2023) in formal reasoning via SFT (Ouyang et al., 2022), test-time scaling (Jaech et al., 2024), and DAPO (Yu et al., 2025a) (*i.e.*, a variant of GRPO that still optimizes Pass@1), aiming to leverage the mixed-symbolic error feedback provided by the compiler for state-based reasoning (Norvig & Intelligence, 2002). However, these provers are rewarded solely based on the correctness of complete proofs as judged by the compiler (Leanprover Community, 2023), making it difficult to identify trajectories containing high-quality intermediate steps that could guide the prover toward solving harder theorems.

To address this, we propose T-RL, an exploration-based training algorithm that augments multitask formal reasoning with Pass@$K$ reinforcement learning (Walder & Karkhanis, 2025; Tang et al., 2025). T-RL maintains consistently higher token-level entropy, which fosters diverse proof strategies and improves overall reasoning ability and reliability. Experimental results demonstrate that Qwen2.5-1.5B and Goedel-Prover-V2 models trained with T-RL outperform their original counterparts on benchmarks such as MiniF2F. Further analysis reveals that T-RL equips these models with improved reasoning performance and enhanced strategy utilization. These findings indicate that T-RL surpasses traditional Pass@1-based approaches in achieving exploration goals. Future research will focus on further validating T-RL's applicability and exploring additional strategies to promote exploration in formal reasoning models.

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

## A   ILLUSTRATIVE EXAMPLES OF FLAWED PROOF REFINEMENT

**Example 2.3 Prompt for revising flawed proofs.**

```lean4
Revise the following incorrect Lean 4 code to make it correct.

```lean4
theorem sum_infinite : Σ' n : ℕ, (1 \/ (4 * n ^ 2 + 16 * n + 7)) = 71 \/ 630 := by
 have h_main : Σ' n : ℕ, (1 \/ (4 * n ^ 2 + 16 * n + 7)) = 0 := by
  apply tsum_eq_single 0
  · simp
  · intro n hn
   have h₁ : 1 \/ (4 * n ^ 2 + 16 * n + 7) = 0 := by
    have h₂ : n ≠ 0 := by
     intro h
     simp_all
    have h₃ : 4 * n ^ 2 + 16 * n + 7 > 1 := by
     rcases n with (_ | _ | n) <;> simp_all [Nat.mul_add, Nat.add_mul, Nat.pow_succ] <;>
     (try omega) <;>
     (try ring_nf at * <;> omega) <;>
     (try nlinarith)
     <;>
     nlinarith
    have h₄ : 1 \/ (4 * n ^ 2 + 16 * n + 7) = 0 := by
     apply Nat.div_eq_of_lt
     <;> omega
    exact h₄
   simp [h₁]
  <;> simp
  <;> aesop

 have h_final : Σ' n : ℕ, (1 \/ (4 * n ^ 2 + 16 * n + 7)) = 71 \/ 630 := by
  rw [h_main]
  <;> norm_num
  <;> rfl

 apply h_final
```

The error messages from Lean 4 are:
[
 {
  "tactic": "apply tsum_eq_single 0",
  "error": "tactic 'apply' failed, failed to unify
   Σ' (b : ?m.7645), ?m.7643 b = ?m.7643 0
with
   Σ' (n : ℕ), 1 \/ (4 * n ^ 2 + 16 * n + 7) = 0
⊢ Σ' (n : ℕ), 1 \/ (4 * n ^ 2 + 16 * n + 7) = 0",
  "severity": "error",
  "line": 7,
  "column": 4,
  "endLine": 7,
  "endColumn": 26
 }
]

Provide a fully corrected Lean 4 code that compiles and is logically correct.
```

## B   CASE STUDIES IN MODEL HACKING AND ERRONEOUS PROOFS

In this subsection, we analyze two typical types of erroneous proofs generated by the models: **(1)** During the refinement stage, the model modifies the statement of the proposition in order to pass the compiler's verification; **(2)** During the testing stage, for problems that are overly difficult, the model repeatedly generates segments of highly similar code.

We note that issue (1) occurs only in the Goedel-Prover-V2 model and is not observed in the Qwen2.5-1.5B model, which may be related to data contamination. As for issue (2), it is likely that the model lacks sufficient confidence to solve the problem, and therefore resorts to repeatedly producing correct yet trivial inferences.

**Appendix B. Model Hacking and Erroneous Proofs**

```
Sometimes prover tends to generate trivial propositions

```lean4
theorem imo_1969_p2 : True := by
  trivial
```

And sometimes the prover may repeatedly generate the same propositions over and over again

```lean4
theorem omni_theorem_2954 {x y : ℤ} (hx : x > 0) (hy : y > 0)
  (hxy : x * y = 24 ∧ x + y = 11) :
  max x y − min x y = 5 := by
 have h_main : x = 3 ∧ y = 8 ∨ x = 8 ∧ y = 3 := by
  have h₁ : x > 0 := hx
  have h₂ : y > 0 := hy
  have h₃ : x * y = 24 := hxy.1
  have h₄ : x + y = 11 := hxy.2
  have h₅ : x ≤ 11 := by
   nlinarith
  have h₆ : y ≤ 11 := by
   nlinarith
  have h₇ : x ≥ 1 := by linarith
  have h₈ : y ≥ 1 := by linarith
  -- We now check all possible values of x and y that satisfy the conditions
  have h₉ : x ≤ 11 := by nlinarith
  have h₁₀ : y ≤ 11 := by nlinarith
  have h₁₁ : x ≥ 1 := by linarith
  have h₁₂ : y ≥ 1 := by linarith
  -- We use the fact that x and y are positive integers and their sum is 11 to find the possible pairs
  have h₁₃ : x = 3 ∧ y = 8 ∨ x = 8 ∧ y = 3 := by
   -- We use the fact that x and y are positive integers and their sum is 11 to find the possible pairs
   have h₁₄ : x ≤ 11 := by nlinarith
   have h₁₅ : y ≤ 11 := by nlinarith
   have h₁₆ : x ≥ 1 := by linarith
   have h₁₇ : y ≥ 1 := by linarith
   -- We use the fact that x and y are positive integers and their sum is 11 to find the possible pairs
   interval_cases x <;> norm_num at h₃ h₄ ⊢ <;>
    (try omega) <;>
    (try
     {
      have h₁₈ : y ≤ 11 := by nlinarith
      have h₁₉ : y ≥ 1 := by linarith
      interval_cases y <;> norm_num at h₃ h₄ ⊢ <;> omega
     }) <;>
    (try omega) <;>
    (try
     {
      aesop
     })
   exact h₁₃

 have h_final : max x y − min x y = 5 := by
  rcases h_main with (⟨rfl, rfl⟩ | ⟨rfl, rfl⟩)
  · -- Case x = 3, y = 8
   simp [max_eq_right, min_eq_left, le_of_lt]
   <;> norm_num
  · -- Case x = 8, y = 3
   simp [max_eq_left, min_eq_right, le_of_lt]
   <;> norm_num
 exact h_final
```
```

## C   REFERENCE FOR TACTIC STATISTICS

In this subsection, we present the complete list of tactics that were statistically analyzed in Section 4.2, as shown in Example C.

The following list summarizes a set of commonly used Lean tactics that frequently appear in proof generation. These tactics cover a variety of proof styles, including algebraic simplification (ring_nf, norm_num, field_simp), case analysis (cases, induction, by_cases),

rewriting (`rw`, `simp`, `unfold`), and structural reasoning (`have`, `apply`, `constructor`, `by_contra`).

> **Appendix C. Tactic List**
>
> ```
> tactic_list = [
>   "have", "rw", "norm_num", "linarith", "nlinarith", "ring_nf", "exact", "calc",
>   "simp", "cases", "induction", "apply", "intro", "constructor", "by_contra", "refine'",
>   "split", "refl", "assumption", "contradiction", "left", "right", "field_simp", "norm_cast",
>   "unfold", "change", "simp only", "generalize", "subst", "by_cases", "cases'", "omega",
>     "interval_cases", "rcases"
> ]
> ```

## D  THE USE OF LARGE LANGUAGE MODELS

We clarify that LLMs were used only in the later stages of writing this paper for proofreading and polishing. LLMs did not contribute any original research content. All authors take full responsibility for the final written content.

