# OpenReview forum: "Advancing Formal Mathematical Reasoning with Explorative Reinforcement Learning"
_ICLR.cc/2026/Conference — Submitted to ICLR 2026_

### Official Review · Reviewer_SbV2 · 2025-10-30

**Soundness:** 4
**Presentation:** 4
**Contribution:** 2
**Rating:** 6
**Confidence:** 4

**Summary:**

The authors apply pass@k aware reinforcement learning to Lean 4 theorem proving. They combine ideas from pass@k policy optimization (PKPO), GRPO, and iVML self-refinement to towards the theorem proving domain and achieve substantial improvements in theorem proving accuracy and efficiency. Moreover they analyze exploration and diversity via types of tactics, tactic length, etc.

**Strengths:**

This paper achieves strong empirical results, surpassing larger models with larger budgets using their method. Their gains hold across several benchmarks and their analysis is fairly thorough, investigating additional metrics that are appropriate for this exploration focused approach.

**Weaknesses:**

- This paper has limited methodological novelty-- the RL objectives are recombined from previous work and the inference strategies are standard.
- The tactic length and tactic types counts are not conditioned on proof success making it a little correlational, can tighten up this analysis slightly

**Questions:**

- Is there a clear explanation for why Prover Bench benefits so strongly from this approach?
- Are the diverse strategies/longer proof lengths corresponding with proof success?

---

### Official Review · Reviewer_xYdd · 2025-10-31

**Soundness:** 2
**Presentation:** 1
**Contribution:** 1
**Rating:** 2
**Confidence:** 3

**Summary:**

The paper studies the problem of training large language models for formal reasoning with reinforcement learning. Most existing approaches in formal reasoning optimize Pass@1, and the paper argues that this results in lower entropy over the course of training and the performance saturation at low k. Similar to other work on natural language reasoning in LLMs, the authors propose training LLMs for theorem proving by directly optimizing a grouped Pass@K reward along with a compiler-based self-refinement auxiliary task. The approach is evaluated empirically with Qwen2.5-1.5B and Goedel-Prover-V2. The empirical results indicate better Pass@32 performance for models trained with the Pass@k objective over models trained with a Pass@1 objective. The authors also introduce two evaluation metrics, Average Reasoning Ability (ARA) and Reliable Reasoning Ability (RRA) to account for Pass@k where the proposed approach outperforms a GRPO baseline. The tactic-level analyses show higher token-entropy, more use of contradiction/induction, and reduced reliance on sorry for models trained with the proposed approach. The authors also couple the trained model with a test-time scaling approach and demonstrate competitive or better MiniF2F results than baselines.

**Strengths:**

* The paper takes the simple idea of optimizing for Pass@k and applies it in the context of training LLM-based theorem provers.
* The proposed approach provides notable performance improvements at Pass@32, across two different model families.
* I also like the idea of combining the model optimized for Pass@k with a matching test-time scaling strategy.

**Weaknesses:**

* One of the major weaknesses of the paper is the quality of writing. Despite the central idea being quite simple (that is not a weakness!) the writing is quite hard to follow. Some examples below:
    * in L135-136, the sentence does not seem complete and does not make sense. The authors refer to iVML but it is not cited, nor mentioned before that point in the paper.
    * L155 states "enable tree search awareness", what is the tree search being referred to here? And the name T-RL is introduced without much context.
    * Section 3 starts directly with some empirical results without any context while the details are in Section 4.
    * The benchmarks used are not cited.
    * The paper also does not provide much in terms of clearn motivations for _why_ you would want Pass@k optimization in formal reasoning.
* The paper talks about the auxilliary self refinement task, but there is no experiment on what, if anything, it contributes or how important it is for test-time scaling.
* The authors describe iVML as "iteratively refine groups of proofs based on error feedback from the previous
iteration" but this seems inconsistent from the iVML paper and the authors do not provide much in terms of details of how exactly it is implemented in their setup.
* There are no ablations in the paper to understand how the method actually works. For instance, how does the training k affect the k that can be used at test time? How low or high can the k go before you see diminishing returns.
* Finally, in the absence of experiments to study how the Pass@k optimization affects formal reasoning in more detail, the novelty of the paper seems limited to simply applying existing techniques.

**Questions:**

* How exactly is iVML implemented in your setup?
* One of the base models you use, Goedel-Prover, already uses verifier based self-refinement. So why is doing that necessary in your setup?

---

### Official Review · Reviewer_N8kv · 2025-10-31

**Soundness:** 2
**Presentation:** 2
**Contribution:** 2
**Rating:** 2
**Confidence:** 3

**Summary:**

This paper introduces T-RL, an exploration-oriented reinforcement learning method for theorem proving. Building on existing frameworks such as GRPO, it replaces the standard Pass@1 objective with a Pass@K-aware reward that encourages exploration across multiple rollouts and integrates an auxiliary self-refinement task. The paper aims to enhance proof diversity and exploration. Results on MiniF2F, FormalMATH-Lite, and ProverBench show modest improvements over the normal Pass@1 reward for most settings.

**Strengths:**

- The proposed method effectively increases the diversity of proof strategies, encouraging the prover to explore a wider range of reasoning patterns.
- Empirical analysis indicates that the model learns to use a richer set of Lean tactics (e.g., by_contra, induction, rcases), showing more human-like exploration of proof space.

**Weaknesses:**

- The paper’s two main ideas—Pass@K reinforcement and iVML sampling—are directly adopted from prior studies. Their adaptation to theorem proving is straightforward and lacks new contributions and insights. The manuscript does not clearly articulate what additional insights arise from applying these existing techniques to theorem proving.

- The contribution of the auxiliary task remains unclear since no ablation study isolates its impact. It is therefore uncertain whether improvements come from the exploration-aware reward or from additional supervision via refinement.

- Exploration benefits are not convincingly tied to performance. While T-RL increases token-level entropy and tactic diversity, the performance gains on MiniF2F and FormalMATH-Lite are marginal (Tables 1 & 2). Only on ProverBench with Goedel-Prover does the method show a notable improvement, which might stem from model or dataset differences rather than the training objective itself.

- The results presentation is hard to follow. Experimental settings are scattered and under-explained, making the empirical claims difficult to interpret.

**Questions:**

- Which specific results best illustrate a clear advantage of the proposed T-RL method over the standard Pass@1 GRPO baseline?

- Why do Goedel-Prover results on the Proverbench show significantly larger improvements than Qwen2.5-1.5B?

- Ablation of the auxiliary task: How much does the self-refinement objective contribute to final performance?

- Line 155 mentions that Equation (6) is adapted “to enable tree-search awareness.” What exactly does tree-search awareness mean in this context? There is no explicit tree-search mechanism in the formulation.

- How are the experiments in Table 1 configured? What does "Multi-Task" exactly mean in this set of experiments?

- How should the sampling-budget notation (e.g., “1 × 32 × 4”) be interpreted. Does it represent the number of samples, inference rounds, or refinement iterations?

---

### Meta-Review · Area_Chair_B1dV · 2025-12-09

**Summary:**

1. All three reviewers expressed concerns about the limited novelty of this work. For example, the two main ideas—Pass@K reinforcement and iVML sampling—are largely adopted from prior studies.
2. The ablation study is insufficient. For example, the contribution of the auxiliary task remains unclear, and it is not explained how the training value of k affects the k used at test time.
3. The writing quality is not satisfactory and needs substantial improvement.

**Reviewer Concerns:**

No rebuttal was submitted.

**Reviewer Scores:**

As no rebuttal was submitted, the reviewers will not revise their scores.

---

### Decision · Program_Chairs · 2026-01-26

Reject